# VEEGAN: Reducing Mode Collapse in GANs using Implicit Variational Learning

**Akash Srivastava**
School of Informatics
University of Edinburgh
akash.srivastava@ed.ac.uk

**Lazar Valkov**
School of Informatics
University of Edinburgh
L.Valkov@sms.ed.ac.uk

**Chris Russell**
The Alan Turing Institute
London
crussell@turing.ac.uk

**Michael U. Gutmann**
School of Informatics
University of Edinburgh
Michael.Gutmann@ed.ac.uk

**Charles Sutton**
School of Informatics & The Alan Turing Institute
University of Edinburgh
csutton@inf.ed.ac.uk

## Abstract

Deep generative models provide powerful tools for distributions over complicated manifolds, such as those of natural images. But many of these methods, including generative adversarial networks (GANs), can be difficult to train, in part because they are prone to mode collapse, which means that they characterize only a few modes of the true distribution. To address this, we introduce VEEGAN, which features a reconstructor network, reversing the action of the generator by mapping from data to noise. Our training objective retains the original asymptotic consistency guarantee of GANs, and can be interpreted as a novel autoencoder loss over the noise. In sharp contrast to a traditional autoencoder over data points, VEEGAN does not require specifying a loss function over the data, but rather only over the representations, which are standard normal by assumption. On an extensive set of synthetic and real world image datasets, VEEGAN indeed resists mode collapsing to a far greater extent than other recent GAN variants, and produces more realistic samples.

## 1   Introduction

Deep generative models are a topic of enormous recent interest, providing a powerful class of tools for the unsupervised learning of probability distributions over difficult manifolds such as natural images [7, 11, 18]. Deep generative models are usually implicit statistical models [3], also called implicit probability distributions, meaning that they do not induce a density function that can be tractably computed, but rather provide a simulation procedure to generate new data points. Generative adversarial networks (GANs) [7] are an attractive such method, which have seen promising recent successes [17, 20, 23]. GANs train two deep networks in concert: a generator network that maps random noise, usually drawn from a multi-variate Gaussian, to data items; and a discriminator network that estimates the likelihood ratio of the generator network to the data distribution, and is trained

using an adversarial principle. Despite an enormous amount of recent work, GANs are notoriously fickle to train, and it has been observed [1, 19] that they often suffer from *mode collapse*, in which the generator network learns how to generate samples from a few modes of the data distribution but misses many other modes, even though samples from the missing modes occur throughout the training data.

To address this problem, we introduce VEEGAN,[1] a variational principle for estimating implicit probability distributions that avoids mode collapse. While the generator network maps Gaussian random noise to data items, VEEGAN introduces an additional *reconstructor* network that maps the true data distribution to Gaussian random noise. We train the generator and reconstructor networks jointly by introducing an implicit variational principle, which encourages the reconstructor network not only to map the data distribution to a Gaussian, but also to approximately reverse the action of the generator. Intuitively, if the reconstructor learns both to map all of the true data to the noise distribution and is an approximate inverse of the generator network, this will encourage the generator network to map from the noise distribution to the entirety of the true data distribution, thus resolving mode collapse.

Unlike other adversarial methods that train reconstructor networks [4, 5, 22], the noise autoencoder dramatically reduces mode collapse. Unlike recent adversarial methods that also make use of a data autoencoder [1, 13, 14], VEEGAN autoencodes noise vectors rather than data items. This is a significant difference, because choosing an autoencoder loss for images is problematic, but for Gaussian noise vectors, an $\ell_2$ loss is entirely natural. Experimentally, on both synthetic and real-world image data sets, we find that VEEGAN is dramatically less susceptible to mode collapse, and produces higher-quality samples, than other state-of-the-art methods.

## 2   Background

Implicit probability distributions are specified by a sampling procedure, but do not have a tractable density [3]. Although a natural choice in many settings, implicit distributions have historically been seen as difficult to estimate. However, recent progress in formulating density estimation as a problem of supervised learning has allowed methods from the classification literature to enable implicit model estimation, both in the general case [6, 10] and for deep generative adversarial networks (GANs) in particular [7]. Let $\{x_i\}_{i=1}^{N}$ denote the training data, where each $x_i \in \mathbb{R}^D$ is drawn from an unknown distribution $p(x)$. A GAN is a neural network $G_\gamma$ that maps representation vectors $z \in \mathbb{R}^K$, typically drawn from a standard normal distribution, to data items $x \in \mathbb{R}^D$. Because this mapping defines an implicit probability distribution, training is accomplished by introducing a second neural network $D_\omega$, called a discriminator, whose goal is to distinguish generator samples from true data samples. The parameters of these networks are estimated by solving the minimax problem

$$\max_{\omega} \min_{\gamma} \mathcal{O}_{\text{GAN}}(\omega, \gamma) := E_z \left[ \log \sigma \left( D_\omega(G_\gamma(z)) \right) \right] + E_x \left[ \log \left( 1 - \sigma \left( D_\omega(x) \right) \right) \right],$$

where $E_z$ indicates an expectation over the standard normal $z$, $E_x$ indicates an expectation over the data distribution $p(x)$, and $\sigma$ denotes the sigmoid function. At the optimum, in the limit of infinite data and arbitrarily powerful networks, we will have $D_\omega = \log q_\gamma(x)/p(x)$, where $q_\gamma$ is the density that is induced by running the network $G_\gamma$ on normally distributed input, and hence that $q_\gamma = p$ [7].

Unfortunately, GANs can be difficult and unstable to train [19]. One common pathology that arises in GAN training is mode collapse, which is when samples from $q_\gamma(x)$ capture only a few of the modes of $p(x)$. An intuition behind why mode collapse occurs is that the only information that the objective function provides about $\gamma$ is mediated by the discriminator network $D_\omega$. For example, if $D_\omega$ is a constant, then $\mathcal{O}_{\text{GAN}}$ is constant with respect to $\gamma$, and so learning the generator is impossible. When this situation occurs in a localized region of input space, for example, when there is a specific type of image that the generator cannot replicate, this can cause mode collapse.

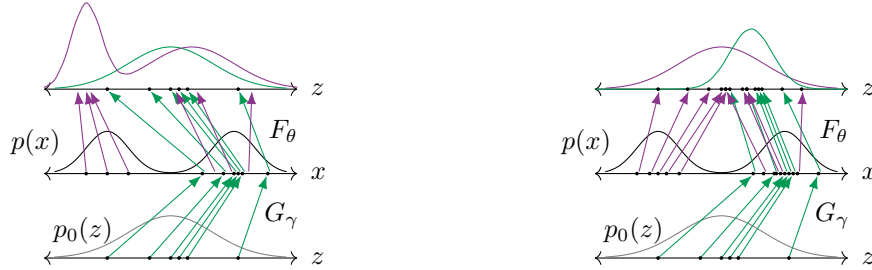

(a) Suppose $F_\theta$ is trained to approximately invert $G_\gamma$. Then applying $F_\theta$ to true data is likely to produce a non-Gaussian distribution, allowing us to detect mode collapse.

(b) When $F_\theta$ is trained to map the data to a Gaussian distribution, then treating $F_\theta \circ G_\gamma$ as an autoencoder provides learning signal to correct $G\gamma$.

Figure 1: Illustration of how a reconstructor network $F_\theta$ can help to detect mode collapse in a deep generative network $G_\gamma$. The data distribution is $p(x)$ and the Gaussian is $p_0(z)$. See text for details.

## 3    Method

The main idea of VEEGAN is to introduce a second network $F_\theta$ that we call the *reconstructor network* which is learned both to map the true data distribution $p(x)$ to a Gaussian and to approximately invert the generator network.

To understand why this might prevent mode collapse, consider the example in Figure 1. In both columns of the figure, the middle vertical panel represents the data space, where in this example the true distribution $p(x)$ is a mixture of two Gaussians. The bottom panel depicts the input to the generator, which is drawn from a standard normal distribution $p_0 = \mathcal{N}(0, I)$, and the top panel depicts the result of applying the reconstructor network to the generated and the true data. The arrows labeled $G_\gamma$ show the action of the generator. The purple arrows labelled $F_\theta$ show the action of the reconstructor on the true data, whereas the green arrows show the action of the reconstructor on data from the generator. In this example, the generator has captured only one of the two modes of $p(x)$. The difference between Figure 1a and 1b is that the reconstructor networks are different.

First, let us suppose (Figure 1a) that we have successfully trained $F_\theta$ so that it is approximately the inverse of $G_\gamma$. As we have assumed mode collapse however, the training data for the reconstructor network $F_\theta$ does not include data items from the "forgotten" mode of $p(x)$, therefore the action of $F_\theta$ on data from that mode is ill-specified. This means that $F_\theta(X), X \sim p(x)$ is unlikely to be Gaussian and we can use this mismatch as an indicator of mode collapse.

Conversely, let us suppose (Figure 1b) that $F_\theta$ is successful at mapping the true data distribution to a Gaussian. In that case, if $G_\gamma$ mode collapses, then $F_\theta$ will not map all $G_\gamma(z)$ back to the original $z$ and the resulting penalty provides us with a strong learning signal for both $\gamma$ and $\theta$.

Therefore, the learning principle for VEEGAN will be to train $F_\theta$ to achieve both of these objectives simultaneously. Another way of stating this intuition is that if the same reconstructor network maps both the true data and the generated data to a Gaussian distribution, then the generated data is likely to coincide with true data. To measure whether $F_\theta$ approximately inverts $G_\gamma$, we use an autoencoder loss. More precisely, we minimize a loss function, like $\ell_2$ loss between $z \sim p_0$ and $F_\theta(G_\gamma(z))$. To quantify whether $F_\theta$ maps the true data distribution to a Gaussian, we use the cross entropy $H(Z, F_\theta(X))$ between $Z$ and $F_\theta(x)$. This boils down to learning $\gamma$ and $\theta$ by minimising the sum of these two objectives, namely

$$\mathcal{O}_{\text{entropy}}(\gamma, \theta) = E\left[\|z - F_\theta(G_\gamma(z))\|_2^2\right] + H(Z, F_\theta(X)). \tag{1}$$

While this objective captures the main idea of our paper, it cannot be easily computed and minimised. We next transform it into a computable version and derive theoretical guarantees.

### 3.1    Objective Function

Let us denote the distribution of the outputs of the reconstructor network when applied to a fixed data item $x$ by $p_\theta(z|x)$ and when applied to all $X \sim p(x)$ by $p_\theta(z) = \int p_\theta(z|x)p(x)\,dx$. The conditional

distribution $p_\theta(z|x)$ is Gaussian with unit variance and, with a slight abuse of notation, (deterministic) mean function $F_\theta(x)$. The entropy term $H(Z, F_\theta(X))$ can thus be written as

$$H(Z, F_\theta(X)) = -\int p_0(z) \log p_\theta(z) dz = -\int p_0(z) \log \int p(x) p_\theta(z|x)\, dx\, dz. \qquad (2)$$

This cross entropy is minimized with respect to $\theta$ when $p_\theta(z) = p_0(z)$ [2]. Unfortunately, the integral on the right-hand side of (2) cannot usually be computed in closed form. We thus introduce a variational distribution $q_\gamma(x|z)$ and by Jensen's inequality, we have

$$-\log p_\theta(z) = -\log \int p_\theta(z|x) p(x) \frac{q_\gamma(x|z)}{q_\gamma(x|z)}\, dx \leq -\int q_\gamma(x|z) \log \frac{p_\theta(z|x)p(x)}{q_\gamma(x|z)}\, dx, \qquad (3)$$

which we use to bound the cross-entropy in (2). In variational inference, strong parametric assumptions are typically made on $q_\gamma$. Importantly, we here relax that assumption, instead representing $q_\gamma$ implicitly as a deep generative model, enabling us to learn very complex distributions. The variational distribution $q_\gamma(x|z)$ plays exactly the same role as the generator in a GAN, and for that reason, we will parameterize $q_\gamma(x|z)$ as the output of a stochastic neural network $G_\gamma(z)$.

In practice minimizing this bound is difficult if $q_\gamma$ is specified implicitly. For instance, it is challenging to train a discriminator network that accurately estimates the unknown likelihood ratio $\log p(x)/q_\gamma(x|z)$, because $q_\gamma(x|z)$, as a conditional distribution, is much more peaked than the joint distribution $p(x)$, making it too easy for a discriminator to tell the two distributions apart. Intuitively, the discriminator in a GAN works well when it is presented a *difficult* pair of distributions to distinguish. To circumvent this problem, we write (see supplementary material)

$$-\int p_0(z) \log p_\theta(z) \leq \mathrm{KL}\left[q_\gamma(x|z) p_0(z) \,\|\, p_\theta(z|x) p(x)\right] - E\left[\log p_0(z)\right]. \qquad (4)$$

Here all expectations are taken with respect to the joint distribution $p_0(z) q_\gamma(x|z)$.

Now, moving to the second term in (1), we define the reconstruction penalty as an expectation of the cost of autoencoding noise vectors, that is, $E\left[d(z, F_\theta(G_\gamma(z)))\right]$. The function $d$ denotes a loss function in representation space $\mathbb{R}^K$, such as $\ell_2$ loss and therefore the term is an autoencoder in representation space. To make this link explicit, we expand the expectation, assuming that we choose $d$ to be $\ell_2$ loss. This yields $E\left[d(z, F_\theta(x))\right] = \int p_0(z) \int q_\gamma(x|z) \|z - F_\theta(x)\|^2\, dx dz$. Unlike a standard autoencoder, however, rather than taking a *data item* as input and attempting to reconstruct it, we autoencode a *representation vector*. This makes a substantial difference in the interpretation and performance of the method, as we discuss in Section 4. For example, notice that we do not include a regularization weight on the autoencoder term in (5), because Proposition 1 below says that this is not needed to recover the data distribution.

Combining these two ideas, we obtain the final objective function

$$\mathcal{O}(\gamma, \theta) = \mathrm{KL}\left[q_\gamma(x|z) p_0(z) \,\|\, p_\theta(z|x) p(x)\right] - E\left[\log p_0(z)\right] + E\left[d(z, F_\theta(x))\right]. \qquad (5)$$

Rather than minimizing the intractable $\mathcal{O}_{\mathrm{entropy}}(\gamma, \theta)$, our goal in VEEGAN is to minimize the upper bound $\mathcal{O}$ with respect to $\gamma$ and $\theta$. Indeed, if the networks $F_\theta$ and $G_\gamma$ are sufficiently powerful, then if we succeed in globally minimizing $\mathcal{O}$, we can guarantee that $q_\gamma$ recovers the true data distribution. This statement is formalized in the following proposition.

**Proposition 1.** *Suppose that there exist parameters $\theta^*, \gamma^*$ such that $\mathcal{O}(\gamma^*, \theta^*) = H[p_0]$, where $H$ denotes Shannon entropy. Then $(\gamma^*, \theta^*)$ minimizes $\mathcal{O}$, and further*

$$p_{\theta^*}(z) := \int p_{\theta^*}(z|x) p(x)\, dx = p_0(z), \quad \text{and} \quad q_{\gamma^*}(x) := \int q_{\gamma^*}(x|z) p_0(z)\, dz = p(x).$$

Because neural networks are universal approximators, the conditions in the proposition can be achieved when the networks $G$ and $F$ are sufficiently powerful.

## 3.2   Learning with Implicit Probability Distributions

This subsection describes how to approximate $\mathcal{O}$ when we have implicit representations for $q_\gamma$ and $p_\theta$ rather than explicit densities. In this case, we cannot optimize $\mathcal{O}$ directly, because the KL divergence

---

**Algorithm 1** VEEGAN training

---

1: **while** not converged **do**
2:     **for** $i \in \{1 \ldots N\}$ **do**
3:         Sample $z^i \sim p_0(z)$
4:         Sample $x_g^i \sim q_\gamma(x|z_i)$
5:         Sample $x^i \sim p(x)$
6:         Sample $z_g^i \sim p_\theta(z_g|x_i)$
7:         $g_\omega \leftarrow -\nabla_\omega \frac{1}{N} \sum_i \log \sigma \left( D_\omega(z^i, x_g^i) \right) + \log \left( 1 - \sigma \left( D_\omega(z_g^i, x^i) \right) \right)$       ▷ Compute $\nabla_\omega \hat{\mathcal{O}}_{\text{LR}}$
8:
9:         $g_\theta \leftarrow \nabla_\theta \frac{1}{N} \sum_i d(z^i, x_g^i)$       ▷ Compute $\nabla_\theta \hat{\mathcal{O}}$
10:
11:         $g_\gamma \leftarrow \nabla_\gamma \frac{1}{N} \sum_i D_\omega(z^i, x_g^i) + \frac{1}{N} \sum_i d(z^i, x_g^i)$       ▷ Compute $\nabla_\gamma \hat{\mathcal{O}}$
12:
13:         $\omega \leftarrow \omega - \eta g_\omega; \theta \leftarrow \theta - \eta g_\theta; \gamma \leftarrow \gamma - \eta g_\gamma$       ▷ Perform SGD updates for $\omega, \theta$ and $\gamma$

---

in (5) depends on a density ratio which is unknown, both because $q_\gamma$ is implicit and also because $p(x)$ is unknown. Following [4, 5], we estimate this ratio using a discriminator network $D_\omega(x, z)$ which we will train to encourage

$$D_\omega(z, x) = \log \frac{q_\gamma(x|z)p_0(z)}{p_\theta(z|x)p(x)}. \tag{6}$$

This will allow us to estimate $\mathcal{O}$ as

$$\hat{\mathcal{O}}(\omega, \gamma, \theta) = \frac{1}{N} \sum_{i=1}^{N} \mathcal{D}_\omega(z^i, x_g^i) + \frac{1}{N} \sum_{i=1}^{N} d(z^i, x_g^i), \tag{7}$$

where $(z^i, x_g^i) \sim p_0(z)q_\gamma(x|z)$. In this equation, note that $x_g^i$ is a function of $\gamma$; although we suppress this in the notation, we do take this dependency into account in the algorithm. We use an auxiliary objective function to estimate $\omega$. As mentioned earlier, we omit the entropy term $-E\left[\log p_0(z)\right]$ from $\hat{\mathcal{O}}$ as it is constant with respect to all parameters. In principle, any method for density ratio estimation could be used here, for example, see [9, 21]. In this work, we will use the logistic regression loss, much as in other methods for deep adversarial training, such as GANs [7], or for noise contrastive estimation [8]. We will train $D_\omega$ to distinguish samples from the joint distribution $q_\gamma(x|z)p_0(z)$ from $p_\theta(z|x)p(x)$. The objective function for this is

$$\mathcal{O}_{\text{LR}}(\omega, \gamma, \theta) = -E_\gamma \left[\log \left(\sigma \left(D_\omega(z, x)\right)\right)\right] - E_\theta \left[\log \left(1 - \sigma \left(D_\omega(z, x)\right)\right)\right], \tag{8}$$

where $E_\gamma$ denotes expectation with respect to the joint distribution $q_\gamma(x|z)p_0(x)$ and $E_\theta$ with respect to $p_\theta(z|x)p(x)$. We write $\hat{\mathcal{O}}_{\text{LR}}$ to indicate the Monte Carlo estimate of $\mathcal{O}_{\text{LR}}$. Our learning algorithm optimizes this pair of equations with respect to $\gamma, \omega, \theta$ using stochastic gradient descent. In particular, the algorithms aim to find a simultaneous solution to $\min_\omega \hat{\mathcal{O}}_{\text{LR}}(\omega, \gamma, \theta)$ and $\min_{\theta, \gamma} \hat{\mathcal{O}}(\omega, \gamma, \theta)$. This training procedure is described in Algorithm 1. When this procedure converges, we will have that $\omega^* = \arg\min_\omega \mathcal{O}_{\text{LR}}(\omega, \gamma^*, \theta^*)$, which means that $D_{\omega^*}$ has converged to the likelihood ratio (6). Therefore $(\gamma^*, \theta^*)$ have also converged to a minimum of $\mathcal{O}$.

We also found that pre-training the reconstructor network on samples from $p(x)$ helps in some cases.

## 4 Relationships to Other Methods

An enormous amount of attention has been devoted recently to improved methods for GAN training, and we compare ourselves to the most closely related work in detail.

**BiGAN/Adversarially Learned Inference** BiGAN [4] and Adversarially Learning Inference (ALI) [5] are two essentially identical recent adversarial methods for learning both a deep generative network $G_\gamma$ and a reconstructor network $F_\theta$. Likelihood-free variational inference (LFVI) [22] extends this idea to a hierarchical Bayesian setting. Like VEEGAN, all of these methods also use a discriminator $D_\omega(z, x)$ on the joint $(z, x)$ space. However, the VEEGAN objective function $\mathcal{O}(\theta, \gamma)$

provides significant benefits over the logistic regression loss over $\theta$ and $\gamma$ that is used in ALI/BiGAN, or the KL-divergence used in LFVI.

In all of these methods, just as in vanilla GANs, the objective function depends on $\theta$ and $\gamma$ only via the output $D_\omega(z, x)$ of the discriminator; therefore, if there is a mode of data space in which $D_\omega$ is insensitive to changes in $\theta$ and $\gamma$, there will be mode collapse. In VEEGAN, by contrast, the reconstruction term does not depend on the discriminator, and so can provide learning signal to $\gamma$ or $\theta$ even when the discriminator is constant. We will show in Section 5 that indeed VEEGAN is dramatically less prone to mode collapse than ALI.

**InfoGAN**   While differently motivated to obtain disentangled representation of the data, InfoGAN also uses a latent-code reconstruction based penalty in its cost function. But unlike VEEGAN, only a part of the latent code is reconstructed in InfoGAN. Thus, InfoGAN is similar to VEEGAN in that it also includes an autoencoder over the latent codes, but the key difference is that InfoGAN does not also train the reconstructor network on the true data distribution. We suggest that this may be the reason that InfoGAN was observed to require some of the same stabilization tricks as vanilla GANs, which are not required for VEEGAN.

**Adversarial Methods for Autoencoders**   A number of other recent methods have been proposed that combine adversarial methods and autoencoders, whether by explicitly regularizing the GAN loss with an autoencoder loss [1, 13], or by alternating optimization between the two losses [14]. In all of these methods, the autoencoder is over images, i.e., they incorporate a loss function of the form $\lambda d(x, G_\gamma(F_\theta(x)))$, where $d$ is a loss function over images, such as pixel-wise $\ell_2$ loss, and $\lambda$ is a regularization constant. Similarly, variational autoencoders [12, 18] also autoencode images rather than noise vectors. Finally, the adversarial variational Bayes (AVB) [15] is an adaptation of VAEs to the case where the posterior distribution $p_\theta(z|x)$ is implicit, but the data distribution $q_\gamma(x|z)$, must be explicit, unlike in our work.

Because these methods autoencode data points, they share a crucial disadvantage. Choosing a good loss function $d$ over natural images can be problematic. For example, it has been commonly observed that minimizing an $\ell_2$ reconstruction loss on images can lead to blurry images. Indeed, if choosing a loss function over images were easy, we could simply train an autoencoder and dispense with adversarial learning entirely. By contrast, in VEEGAN we autoencode the noise vectors $z$, and *choosing a good loss function for a noise autoencoder is easy*. The noise vectors $z$ are drawn from a standard normal distribution, using an $\ell_2$ loss on $z$ is entirely natural — and does not, as we will show in Section 5, result in blurry images compared to purely adversarial methods.

## 5   Experiments

Quantitative evaluation of GANs is problematic because implicit distributions do not have a tractable likelihood term to quantify generative accuracy. Quantifying mode collapsing is also not straightforward, except in the case of synthetic data with known modes. For this reason, several indirect metrics have recently been proposed to evaluate GANs specifically for their mode collapsing behavior [1, 16]. However, none of these metrics are reliable on their own and therefore we need to compare across a number of different methods. Therefore in this section we evaluate VEEGAN on several synthetic and real datasets and compare its performance against vanilla GANs [7], Unrolled GAN [16] and ALI [5] on five different metrics. Our results strongly suggest that VEEGAN does indeed resolve mode collapse in GANs to a large extent. Generally, we found that VEEGAN performed well with default hyperparameter values, so we did not tune these. Full details are provided in the supplementary material.

### 5.1   Synthetic Dataset

Mode collapse can be accurately measured on synthetic datasets, since the true distribution and its modes are known. In this section we compare all four competing methods on three synthetic datasets of increasing difficulty: a mixture of eight 2D Gaussian distributions arranged in a ring, a mixture of twenty-five 2D Gaussian distributions arranged in a grid [2] and a mixture of ten 700 dimensional

Table 1: Sample quality and degree of mode collapse on mixtures of Gaussians. VEEGAN consistently captures the highest number of modes and produces better samples.

| | 2D Ring | | 2D Grid | | 1200D Synthetic | |
|---|---|---|---|---|---|---|
| | Modes (Max 8) | % High Quality Samples | Modes (Max 25) | % High Quality Samples | Modes (Max 10) | % High Quality Samples |
| **GAN** | 1 | 99.3 | 3.3 | 0.5 | 1.6 | 2.0 |
| **ALI** | 2.8 | 0.13 | 15.8 | 1.6 | 3 | 5.4 |
| **Unrolled GAN** | 7.6 | 35.6 | 23.6 | 16 | 0 | 0.0 |
| **VEEGAN** | **8** | **52.9** | **24.6** | **40** | **5.5** | **28.29** |

Gaussian distributions embedded in a 1200 dimensional space. This mixture arrangement was chosen to mimic the higher dimensional manifolds of natural images. All of the mixture components were isotropic Gaussians. For a fair comparison of the different learning methods for GANs, we use the same network architectures for the reconstructors and the generators for all methods, namely, fully-connected MLPs with two hidden layers. For the discriminator we use a two layer MLP without dropout or normalization layers. VEEGAN method works for both deterministic and stochastic generator networks. To allow for the generator to be a stochastic map we add an extra dimension of noise to the generator input that is not reconstructed.

To quantify the mode collapsing behavior we report two metrics: We sample points from the generator network, and count a sample as *high quality*, if it is within three standard deviations of the nearest mode, for the 2D dataset, or within 10 standard deviations of the nearest mode, for the 1200D dataset. Then, we report the *number of modes captured* as the number of mixture components whose mean is nearest to at least one high quality sample. We also report the percentage of high quality samples as a measure of sample quality. We generate 2500 samples from each trained model and average the numbers over five runs. For the unrolled GAN, we set the number of unrolling steps to five as suggested in the authors' reference implementation.

As shown in Table 1, VEEGAN captures the greatest number of modes on all the synthetic datasets, while consistently generating higher quality samples. This is visually apparent in Figure 2, which plot the generator distributions for each method; the generators learned by VEEGAN are sharper and closer to the true distribution. This figure also shows why it is important to measure sample quality and mode collapse simultaneously, as either alone can be misleading. For instance, the GAN on the 2D ring has 99.3% sample quality, but this is simply because the GAN collapses all of its samples onto one mode (Figure 2b). On the other extreme, the unrolled GAN on the 2D grid captures almost all the modes in the true distribution, but this is simply because that it is generating highly dispersed samples (Figure 2i) that do not accurately represent the true distribution, hence the low sample quality. All methods had approximately the same running time, except for unrolled GAN, which is a few orders of magnitude slower due to the unrolling overhead.

## 5.2 Stacked MNIST

Following [16], we evaluate our methods on the stacked MNIST dataset, a variant of the MNIST data specifically designed to increase the number of discrete modes. The data is synthesized by stacking three randomly sampled MNIST digits along the color channel resulting in a 28x28x3 image. We now expect 1000 modes in this data set, corresponding to the number of possible triples of digits.

Again, to focus the evaluation on the difference in the learning algorithms, we use the same generator architecture for all methods. In particular, the generator architecture is an off-the-shelf standard implementation[3] of DCGAN [17].

For Unrolled GAN, we used a standard implementation of the DCGAN discriminator network. For ALI and VEEGAN, the discriminator architecture is described in the supplementary material. For the reconstructor in ALI and VEEGAN, we use a simple two-layer MLP for the reconstructor without any regularization layers.

|  | Stacked-MNIST | | CIFAR-10 |
|---|---|---|---|
|  | Modes (Max 1000) | KL | IvOM |
| **DCGAN** | 99 | 3.4 | $0.00844 \pm 0.002$ |
| **ALI** | 16 | 5.4 | $\mathbf{0.0067} \pm 0.004$ |
| **Unrolled GAN** | 48.7 | 4.32 | $0.013 \pm 0.0009$ |
| **VEEGAN** | **150** | **2.95** | $\mathbf{0.0068} \pm 0.0001$ |

Table 2: Degree of mode collapse, measured by modes captured and the inference via optimization measure (IvOM), and sample quality (as measured by KL) on Stacked-MNIST and CIFAR. VEEGAN captures the most modes and also achieves the highest quality.

Finally, for VEEGAN we pretrain the reconstructor by taking a few stochastic gradient steps with respect to $\theta$ before running Algorithm 1. For all methods other than VEEGAN, we use the enhanced generator loss function suggested in [7], since we were not able to get sufficient learning signals for the generator without it. VEEGAN did not require this adjustment for successful training.

As the true locations of the modes in this data are unknown, the number of modes are estimated using a trained classifier as described originally in [1]. We used a total of 26000 samples for all the models and the results are averaged over five runs. As a measure of quality, following [16] again, we also report the KL divergence between the generator distribution and the data distribution. As reported in Table 2, VEEGAN not only captures the most modes, it consistently matches the data distribution more closely than any other method. Generated samples from each of the models are shown in the supplementary material.

## 5.3 CIFAR

Finally, we evaluate the learning methods on the CIFAR-10 dataset, a well-studied and diverse dataset of natural images. We use the same discriminator, generator, and reconstructor architectures as in the previous section. However, the previous mode collapsing metric is inappropriate here, owing to CIFAR's greater diversity. Even within one of the 10 classes of CIFAR, the intra-group diversity is very high compared to any of the 10 classes of MNIST. Therefore, for CIFAR it is inappropriate to assume, as the metrics of the previous subsection do, that each labelled class corresponds to a single mode of the data distribution.

Instead, we use a metric introduced by [16] which we will call the inference via optimization metric (IvOM). The idea behind this metric is to compare real images from the test set to the nearest generated image; if the generator suffers from mode collapse, then there will be some images for which this distance is large. To quantify this, we sample a real image $x$ from the test set, and find the closest image that the GAN is capable of generating, i.e. optimizing the $\ell_2$ loss between $x$ and generated image $G_\gamma(z)$ with respect to $z$. If a method consistently attains low MSE, then it can be assumed to be capturing more modes than the ones which attain a higher MSE. As before, this metric can still be fooled by highly dispersed generator distributions, and also the $\ell_2$ metric may favour generators that produce blurry images. Therefore we will also evaluate sample quality visually. All numerical results have been averaged over five runs. Finally, to evaluate whether the noise autoencoder in VEEGAN is indeed superior to a more traditional data autoencoder, we compare to a variant, which we call VEEGAN +DAE, that uses a data autoencoder instead, by simply replacing $d(z, F_\theta(x))$ in $\mathcal{O}$ with a data loss $\|x - G_\gamma(F_\theta(x)))\|_2^2$.

As shown in Table 2, ALI and VEEGAN achieve the best IvOM. Qualitatively, however, generated samples from VEEGAN seem better than other methods. In particular, the samples from VEEGAN +DAE are meaningless. Generated samples from VEEGAN are shown in Figure 3b; samples from other methods are shown in the supplementary material. As another illustration of this, Figure 3 illustrates the IvOM metric, by showing the nearest neighbors to real images that each of the GANs were able to generate; in general, the nearest neighbors will be more semantically meaningful than randomly generated images. We omit VEEGAN +DAE from this table because it did not produce plausible images. Across the methods, we see in Figure 3 that VEEGAN captures small details, such as the face of the poodle, that other methods miss.

Figure 2: Density plots of the true data and generator distributions from different GAN methods trained on mixtures of Gaussians arranged in a ring (top) or a grid (bottom).

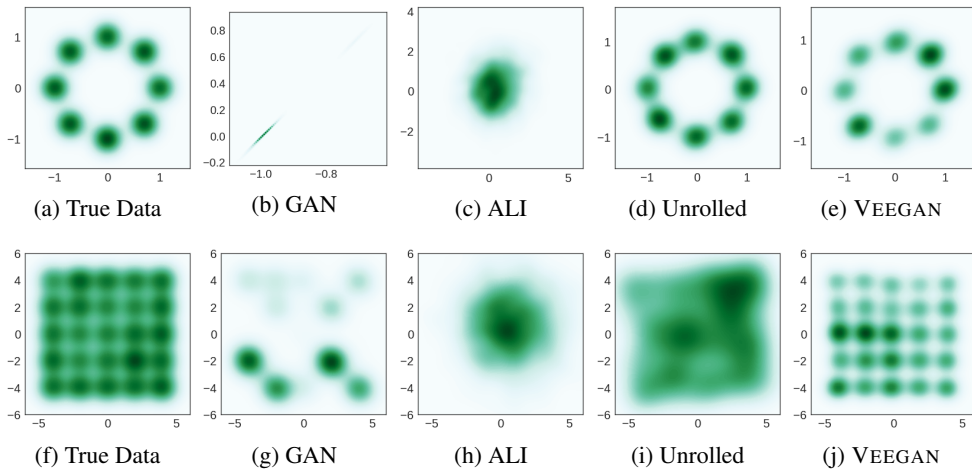

(a) True Data    (b) GAN    (c) ALI    (d) Unrolled    (e) VEEGAN

(f) True Data    (g) GAN    (h) ALI    (i) Unrolled    (j) VEEGAN

Figure 3: Sample images from GANs trained on CIFAR-10. Best viewed magnified on screen.

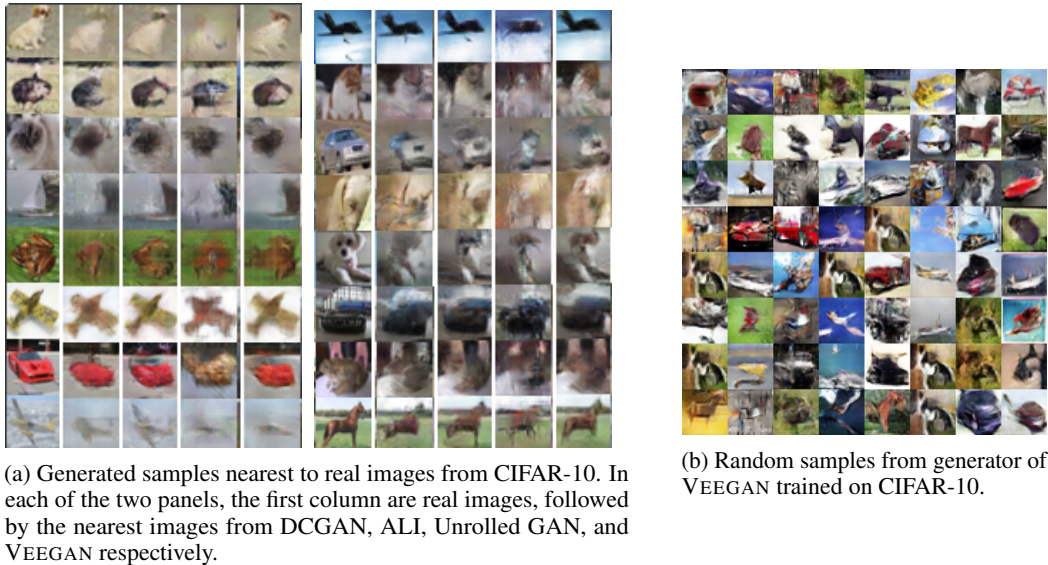

(a) Generated samples nearest to real images from CIFAR-10. In each of the two panels, the first column are real images, followed by the nearest images from DCGAN, ALI, Unrolled GAN, and VEEGAN respectively.

(b) Random samples from generator of VEEGAN trained on CIFAR-10.

## 6  Conclusion

We have presented VEEGAN, a new training principle for GANs that combines a KL divergence in the joint space of representation and data points with an autoencoder over the representation space, motivated by a variational argument. Experimental results on synthetic data and real images show that our approach is much more effective than several state-of-the art GAN methods at avoiding mode collapse while still generating good quality samples.

## Acknowledgement

We thank Martin Arjovsky, Nicolas Collignon, Luke Metz, Casper Kaae Sønderby, Lucas Theis, Soumith Chintala, Stanisław Jastrzębski, Harrison Edwards, Amos Storkey and Paulina Grnarova for their helpful comments. We would like to specially thank Ferenc Huszár for insightful discussions and feedback.

## Footnotes

[1] VEEGAN is a Variational Encoder Enhancement to Generative Adversarial Networks. `https://akashgit.github.io/VEEGAN/`

[2]Experiment follows [5]. Please note that for certain settings of parameters, vanilla GAN can also recover all 25 modes, as was pointed out to us by Paulina Grnarova.

[3]`https://github.com/carpedm20/DCGAN-tensorflow`

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
