[Supplementary Material · veegan-nips-sup.pdf]

# Supplemental Material for VEEGAN: Reducing Mode Collapse in GANs using Implicit Variational Learning

**Akash Srivastava**
School of Informatics
University of Edinburgh
akash.srivastava@ed.ac.uk

**Lazar Valkov**
School of Informatics
University of Edinburgh
L.Valkov@sms.ed.ac.uk

**Chris Russell**
The Alan Turing Institute
London
crussell@turing.ac.uk

**Michael Gutmann**
School of Informatics
University of Edinburgh
Michael.Gutmann@ed.ac.uk

**Charles Sutton**
School of Informatics & The Alan Turing Institute
University of Edinburgh
csutton@inf.ed.ac.uk

## A  Proof of Lower Bound

This appendix completes the proof of the bound in the text that

$$-\int p_0(z) \log p_\theta(z) \le \mathrm{KL}\left[q_\gamma(x|z)p_0(z) \,\|\, p_\theta(z|x)p(x)\right] - E\left[\log p_0(z)\right] \tag{1}$$

where $p_0$ is the standard normal density, and $p_\theta(z) = \int p_\theta(z|x)p(x)\,dx$. As described in the text, introducing a a variational distribution $q_\gamma(x|z)$ yields

$$-\int p_0(z) \log p_\theta(z)\,dz \le -\iint p_0(z)q_\gamma(x|z) \log \frac{p_\theta(z|x)p(x)}{q_\gamma(x|z)}\,dx\,dz. \tag{2}$$

Starting from (2), we obtain a new upper bound by adding a trivial KL divergence to the right hand side of the above inequality

$$
\begin{aligned}
-\int p_0(z) \log p_\theta(z)\,dz &\le -\iint p_0(z)q_\gamma(x|z) \log \frac{p_\theta(z|x)p(x)}{q_\gamma(x|z)}\,dx\,dz \\
&= \iint p_0(z)q_\gamma(x|z) \log \frac{q_\gamma(x|z)}{p_\theta(z|x)p(x)}\,dx\,dz + \int p_0(z) \log \frac{p_0(z)}{p_0(z)}\,dz
\end{aligned} \tag{3}
$$

Now for the upper term in the KL, we have that

$$\int p_0(z) \log p_0(z)\,dz = \int p_0(z) \log p_0(z) \left(\int q_\gamma(x|z)\,dx\right) dz = \iint p_0(z)q_\gamma(x|z) \log p_0(z)\,dx\,dz.$$

Combining with (3) yields

$$
\begin{aligned}
H(Z, F_\theta(X)) \leq & \iint p_0(z)q_\gamma(x|z) \log \frac{q_\gamma(x|z)}{p_\theta(z|x)p(x)} \, dx \, dz + \iint p_0(z)q_\gamma(x|z) \log p_0(z) \, dx \, dz \\
& - \int p_0(z) \log p_0(z) \, dz \\
= & \iint p_0(z)q_\gamma(x|z) \log \frac{q_\gamma(x|z)p_0(z)}{p_\theta(z|x)p(x)} \, dx \, dz - \int p_0(z) \log p_0(z) \, dz \\
= & \text{KL} \left[ q_\gamma(x|z)p_0(z) \, \| \, p_\theta(z|x)p(x) \right] - \int p_0(z) \log p_0(z) \, dz,
\end{aligned}
$$

which completes the proof.

## B    Proof of Proposition 1

**Proposition 1.** *Suppose that there exist parameters $\theta^*, \gamma^*$ such that $\mathcal{O}(\gamma^*, \theta^*) = H[p_0]$, where $H$ denotes Shannon entropy. Then $(\gamma^*, \theta^*)$ minimizes $\mathcal{O}$, and we further have that*

$$
p_{\theta^*}(z) := \int p_{\theta^*}(z|x)p(x) \, dx = p_0(z)
$$

$$
q_{\gamma^*}(x) := \int q_{\gamma^*}(x|z)p_0(z) \, dz = p(x).
$$

*Proof.*  From information theory, we know that $\text{KL} \left[ q_\gamma(x|z)p_0(z) \, \| \, p_\theta(z|x)p(x) \right] \geq 0$. Additionally, we have that $E \left[ d(z, F_\theta(x)) \right] \geq 0,$. Moreover, by definition of $E \, [\,]$ in the proposition,

$$
\begin{aligned}
-E \left[ \log p_0(z) \right] = & - \iint p_0(z)q_\gamma(x|z) \log p_0(z) \, dz dx = - \int p_0(z) \log p_0(z) \, dz \int q_\gamma(x|z) \, dx \\
= & - \int p_0(z) \log p_0(z) \, dz,
\end{aligned}
$$

which is the definition of the Shannon entropy $H[p_0]$ of $p_0$.

This implies that

$$
\begin{aligned}
\mathcal{O}(\gamma, \theta) = & \text{KL} \left[ q_\gamma(x|z)p_0(z) \, \| \, p_\theta(z|x)p(x) \right] - E \left[ \log p_0(z) \right] + E \left[ d(z, F_\theta(x)) \right] \\
\geq & -E \left[ \log p_0(z) \right] \\
= & H[p_0].
\end{aligned}
$$

This bound is attained with equality when $q_\gamma(x|z)p_0(z) = p_\theta(z|x)p(x)$, and when $F_\theta$ inverts $G_\gamma$ on the data distribution, i.e., when $F_\theta(G_\gamma(z)) = z$ for all $z$. (Note that this statement does not require $G$ to be invertible outside of its range.)

Now, if $\mathcal{O}(\gamma^*, \theta^*) = H[p_0]$, subtracting the entropy from both sides implies that $\text{KL} \left[ q_\gamma(x|z)p_0(z) \, \| \, p_\theta(z|x)p(x) \right] = 0$. Because the optimum of the KL divergence is unique, we then have that $q_{\gamma^*}(x|z)p_0(z) = p_{\theta^*}(z|x)p(x)$.

Integrating both sides over $x$ yields the first equality in the proposition, and integrating over $z$ yields the second. $\square$

## C    Discriminator Architecture for ALI and VEEGAN

When using ALI and VEEGAN, the original DCGAN discriminator needs to be augmented in order allow it to operate on pairs of images and noise vectors. In order to achieve this, we flatten the final convolutional layer of DCGAN's discriminator and concatenate it with the input noise vector. Afterwards, we run the concatenation through a hidden layer, and then compute $D_\omega(z, x)$ through a linear transformation.

Table 1: ALI and VEEGAN Discriminator Architecture.

| Operation | #Output | BN? | Activation |
|---|---|---|---|
| $D_\omega(x)$ | | | |
| Conv | 64 | False | Leaky ReLU |
| Conv | 128 | True | Leaky ReLU |
| Conv | 256 | True | Leaky ReLU |
| Conv | 512 | True | Leaky ReLU |
| Flatten | - | - | - |
| $\sigma(D_\omega(z,x))$ | Concatenate $D_\omega(x)$ and $z$ along the first axis. | | |
| Fully Connected | 512 | False | Leaky ReLU |
| Fully Connected | 1 | False | Sigmoid |

Figure 1: VEEGAN method can be used like ALI to perform inference. The means output from the reconstructor network for the real images in the top row are used as the latent features to samples the generated images in the bottom row.

## D  Inference

While not the focus of this work, our method can also be used for inference as in the case of ALI and BiGAN models. Figure 1 shows an example of inference on MNIST. The top row samples are from the dataset. We extract the latent representation vector for each of the real images by running them through the trained reconstructor and then use the resulting vector in the generator to get the generated samples shown in the bottom row of the figure.

## E  Adversarial Methods for Autoencoders

In order to quantify contrast the effect of autoencoding of noise in VEEGAN with autoencoding of data in DAE methods [1, 3] we train DAE version of VEEGAN by simply using the reconstructor network as an inference network. As mentioned before, careful tuning of the weighing parameter $\lambda$ is needed to ensure that the $\ell_2$ loss is only working as a regularizer. Therefore, we run a parameter sweep for $\lambda$. As shown in figure 2 we were not able to obtain any meaningful images for any of the tested values.

Figure 2: CIFAR 10 samples from GANs with data Autoencoders. We did a parameter sweep over the value of $\lambda$ but were unable to generate any meaningful images for any of the values. Figure 2d is generated entirely from the $\ell_2$ loss.

(a) $\lambda = 0.007$     (b) $\lambda = 0.01$     (c) $\lambda = 0.05$     (d) Only $\ell_2$

# F    Stacked MNIST Qualitative Results

Qualitative results from the Stacked MNIST dataset for all the 4 methods.

Figure 3: Samples from trained models for Stacked MNIST dataset.

(a) True Data    (b) DCGAN    (c) ALI    (d) Unrolled    (e) VEEGAN

# G    CelebA Random Sample from ALI and VEEGAN

Additionally, we compared ALI and VEEGAN models on the much bigger CelebA dataset [4] of faces. Our goal is to test how robust each method is when used without extensive tuning of model architecture and hyperparameters on a new dataset. Therefore we use the same model architectures and hyperparameters as we did on the CIFAR-10 data. While ALI failed to produce any meaningful images, VEEGAN generates high quality images of faces. Please note that this does not mean that ALI fails on CelebA in general. Indeed, as [2] show, given higher capacity reconstructor and discriminator with the right hyperparameters, it is possible to generate good quality images on this dataset. Rather, this experiment only suggests that for the simple network that we use for Stacked MNIST and CIFAR experiments, VEEGAN learning method was able to produce reasonable images without any further tuning or hyper parameter search.

Figure 4: ALI on CelebA with simple DCGAN architecture and without tweaking of hyperparameters.

Figure 5: VEEGAN on CelebA with simple DCGAN architecture and default hyperparameters.

# H   CIFAR 10 Random Sample from VEEGAN

Randomly generated samples for CIFAR 10 dataset for all the 4 methods.

Figure 6: DCGAN on CIFAR 10 Dataset

Figure 7: ALI on CIFAR 10 Dataset

Figure 8: Unrolled GAN on CIFAR 10 Dataset

Figure 9: VEEGAN on CIFAR 10 Dataset