[Reviews · NeurIPS 2017]

Reviewer 1



This paper adds a reconstructor netowrk to reverse the action of the generator by mapping from data to noise, to solve the mode collapse problem of gan training. It obtains this reconstruction network by introducing an implicit variational principle to get an upper bound on the cross entropy between the reconstructor network of F_theta(X) and the original noise distribution of z. The paper also shows empirical results on MNIST, CIFAR10 and a synthetic dataset. I feel some parts of the paper are not well written and kind of misleading. Below are my concerns about this paper, I will consider the authors' response to adjust the final score if necessary. 1.The algorithm does not only add a reconstructor network, but also changes the input to the discriminator from only x to (x,z) joint pairs, which I think is a fairly important point and should be mentioned at the begining. 2.Should the d(z^i,x_g^i) be d(z^i,F_theta(x_g^i) ) in all the paper after eq3? 3.Since the proposed model uses the reconstruction on random noise z in the loss function explicitly, should the model eventually choose an F_theta as picture (b) in Figure 1, which perfectly reconstructs z, but is called by the paper as ‘a poor choice’? 4.The idea of reconstructing z is not new, which has been mentinoed in the infoGAN[1] paper. 5.I’ll be interested to see the results of other measurement such as Inception Score[2] or Mode Score[3], which are commanly used in some other papers. And it would be better if the results of experiments on larger datasets such as ImageNet can be added. Reference: [1] InfoGAN: Interpretable Representation Learning by Information Maximizing Generative Adversarial Nets, https://arxiv.org/abs/1606.03657 [2]Improved Techniques for Training GANs,https://arxiv.org/pdf/1606.03498.pdf [3]MODE REGULARIZED GENERATIVE ADVERSARIAL NETWORKS, https://arxiv.org/pdf/1612.02136.pdf I have read the authors' rebuttal, which resolves most of my concerns. I am willing to raise my score to 5.

Reviewer 2



GANs have received a lot of attention lately. They are very good at generating sharp samples in an unsupervised way. However this good sample generation performance comes with a cost: the optimization procedure can be unstable and might lead to mode collapsing -- which happens when the generator can only generate samples that lie in a subspace of the data manifold. There has been some works about providing solutions to this mode collapsing issue. This paper lies within that line of research. Several remarks on the paper below: 1. The paper is very well written and well motivated. I also liked the intuitive explanation of the mode collapsing issue of GANs and the importance of the choice of the reconstructor network. 2. What the reconstructor network is doing is learning the inverse map that the generator is learning. It learns to map data to the latent space. It would be interesting to discuss this in the framework of approximate posterior inference when the conditional distribution defined by the reconstructor network is not deterministic. 3. Minor comment: it was a little bit unsettling to call q(x | z) a variational distribution as this is generally q(z | x). 2. The comparison to existing related works is very well done. Overall a great paper.

Reviewer 3



This paper proposes a variation of the GAN algorithm aimed at improving the mode coverage of the generative distribution. The main idea is to use three networks: a generator, an encoder and a discriminator where the discriminator takes pairs (x,z) as input. The criterion to optimize is a GAN style criterion for the discriminator and a reconstruction in latent space for the encoder (unlike auto-encoders where the reconstruction is in input space). The proposed criterion is properly motivated and experimental evidence is provided to support the claim that the new criterion allows to produce good quality samples (similarly to other GAN approaches) but with more diversity and coverage of the target distribution (i.e. more modes are covered) than other approaches. The results are fairly convincing and the algorithm is evaluated with several of the metrics used in previous literature on missing modes in generative models. However, I believe that the paper could be improved on two aspects: - the criterion is only partially justified by the proposed derivation starting from the cross entropy in Z space and there is an extra distance term that is added without clear justification (it's ok to add terms, but making it sound like it falls out of the derivation is not great) - the connection to other existing approaches, and in particular ALI could be further detailed and the differences could be better explained. More specifically, regarding section 3.1 and the derivation of the objective, I think the exposition is somewhat obfuscating what's really going on: Indeed, I would suggest to rewrite (3) as the following simple lemma: H(Z,F_\theta(X)) <= KL(q_\gamma(x|z)p_0(z) \| p_\theta(z|x)p(x)) - E log p_0(z) which directly follows from combining (1) and (2) And then the authors could argue that they add this extra expected distance term (third term of (3)). But really, this term is not justified by any of the derivation. However, it is important to notice that this term vanishes exactly when the KL term is minimized since the minimum of the KL term is reached when q_\gamma inverses p_\theta i.e. when F_\theta is the inverse of G_\gamma in which case z=F_\theta(G_\gamma(z)). There is also the question of whether this term is necessary. Of course, it helps training the encoder to invert the decoder but it seems possible to optimize only the first part of (5) with respect to both theta and gamma Regarding the connection to ALI: the discriminator is similar to the one in ALI and trained with the same objective. However what is different is the way the discriminator is used when training the generator and "encoder" networks. Indeed, in ALI they are trained by optimizing the JS divergence, while here the generator is trained by optimizing the (estimate of) KL divergence plus the additional reconstruction term and the encoder is purely trained using the reconstruction term. It would be interesting to see what is the effect of varying the weight of the reconstruction term. The authors say that there is no need to add a weight (indeed, the reconstruction term is zero when the KL term is zero as discussed above), but that doesn't mean it cannot have an effect in practice. Side note: there is inconsistency in the notations as d is a distance in latent space, so d(z^i, x_g^i) is not correct and should be replaced by d(z^i, F_\theta(x_g^i)) everywhere. Overall, I feel that this paper presents an interesting new algorithm for training a generative model and the experiments are showing that it does have the claimed effect of improving the mode coverage of other comparable algorithms, but the derivation and justification of the algorithm as well as the comparison to other algorithms could be improved in order to provide better intuition of why this would work.

Reviewer 4



Algorithm: Bound * The introduced bound is very loose, with the distance function not being derived from a bound perspective (the main text of the paper is confusing from this perspective and misleads the reader to believe that the entire loss function can be derived from the variational lower bound). The variational perspective helps justify the BIGAN / ALI loss, not the additional loss which the authors claim to be a key contribution. * Algorithm 1 does not mention the pretraining steps done by the authors in all their experiments. Metrics: * None of the proposed metrics addresses overfitting. VEEGAN could be better at memorizing the training set. IvOM would be best for a model that has memorized the training data. There is no measure of overfitting in the entire paper. Experiments: * Would be nice to have the Inception score measurements on CIFAR-10, to be able to compare to other papers which report this measure. * The explanation that l2 does not work is not very convincing, a lot of papers use l2 reconstruction loss on data with success (CycleGAN is one of them for example), or as cited in the paper [1, 12, 13]. So we would expect to get some reasonable results with this approach. Paper not reproducible: * there are no details in the paper regarding used learning rates or the optimizer used. This makes the paper not reproducible. * By changing the generator loss function for VEEGAN compared to the other methods (line 242 in the paper), the results are not clearly comparable. The authors should have presented baseline with VEEGAN both using the enhanced version of the loss and the zero sum version of the loss. Moreover, line 241 shows that VEEGAN is pretrained with a couple of reconstruction loss iterations before training using the GAN loss. The authors fail to explain why this is needed, and the impact this has on the results, as well as what ‘a few’ means (another aspect that makes the paper not reproducible). A baseline experiment here would have also been useful. Credit: * AGE (http://sites.skoltech.ru/app/data/uploads/sites/25/2017/06/AGE.pdf) also has reconstruction loss in the code space * PPGN trains an autoencoder in the code space (https://arxiv.org/abs/1612.00005), and they use that to learn a prior distribution over codes - instead of fixing the prior * When discussing methods for density ratio estimation (line 145), this paper should be cited: https://arxiv.org/abs/1610.03483, as it discusses deeper connections between density ratio estimation and implicit models. Paper clarity: * Figure 1 is slightly confusing. There is no mention of matching the prior yet, the focus so far in the paper has been about reconstructing data points. Figure 1b) shows a poor choice of F, because the data points are more spread out, hence matching the prior when there is mode collapse in the data space. However, if we look at the pictures from a code reconstruction point of view, Figure 1b) does a good job, while Figure 1a) does not. The two figures are showing the opposite of what the caption says if we are thinking of having an F good at reconstructing codes (which is part of the objective function and emphasized as the key contribution of the paper). * Formulas: not denoting the distribution with respect to which expectations are taken makes it harder for readers to look at the paper (especially in equation 6) * Typo on line 59: the denominator and numerator in the fraction need to be swapped